# Study on the Characteristics and Influence Factor of Methane and Coal Dust Gas/Solid Two-Phase Mixture Explosions

**Yue Wang [1], Zhi Wang [2], Xingyan Cao [2],\* and Haoyue Wei [2]**

1    Key Laboratory of Coal Resources and Green Mining in Xinjiang, Ministry of Education,
Xinjiang Institute of Engineering, Urumqi 830023, China
2    College of Safety Science and Engineering, Nanjing Tech University, Nanjing 210009, China
\*    Correspondence: caoxingyan_007@163.com

**Abstract:** This research aimed to the characteristics and influence factor of methane and coal dust gas/solid two-phase mixture explosions by experiment. Through comparative analysis of flame propagation characteristics, pressure, flame temperature and products, the characteristics of gas/solid explosions and its influence factor were analyzed. And the influence mechanism was also revealed. Results indicate that the coal dust parameter and methane concentration were the important influence factor on mixture explosions. Explosion intensity could be indirectly affected by influencing the flame propagation. Under the determined coal dust parameter, the explosion parameter showed a change trend of increase firstly and then decrease as the methane concentration increased. And it was the greatest at 6% methane concentration. However, the concentration of coal dust corresponding to the maximum pressure was variable and was decreased successively as the methane concentration increased. The corresponding dust concentrations were 500 $g/m^3$ and 200 $g/m^3$ under 2% and 10% methane concentrations, respectively. Meanwhile, the pressure all presented an increasing trend with the reduction of coal dust diameter under five coal dust concentrations, and the explosion intensity was the greatest at 300 $g/m^3$ coal dust concentration. For 2% methane concentration, the explosion would not occur as the dust concentration was less than 400 $g/m^3$. And the same phenomena also appeared as the methane concentration exceeded 10%. The explosion parameter presented the same change trend with the changes of methane concentration and coal dust parameters. Besides, the thermal stability and decomposition oxidation characteristics of burned coal dust were evidently changed compared with unburned coal dust. The weight loss rate and oxidation reaction rate were decreased, and the corresponding temperature was increased. It indicates that coal dust participated in gas/dust two-phase explosion reactions, and the pyrolysis reaction of volatile matter led to an obvious reduction in the weight loss and oxidation reaction rate. And the precipitation of volatile matter also resulted in an obvious pore structure on its surface. The physical parameters and internal components of coal dust were important factors affecting the reaction rates of gas/dust mixture explosions.

**Keywords:** gas/dust two phases; explosion characteristics; influence factor; flame temperature; product analysis

## 1. Induction

The explosion accidents have become more frequent and severe with the expeditious development of economy and ceaseless progress of social, and it is more prone to a compound explosion consisting of two combustible substances, gas and dust [1–3]. The mechanism of gas/dust two-phase mixture explosions become more complex and the explosion power is greater, seriously causing the enormous economic loss and greatly restricting the national economy development [4,5]. In view of the industrial disaster,

the explosion characteristics of gas/dust two-phase mixture should be researched in-depth [6–9]. Further, the effective protective method can be taken to avoid derivative accidents and disasters [10–13].

Relevant scholars have conducted the study on the characteristics of gas/dust two-phase mixture explosions. Kundu [14] proposed that the two-phase mixtures were easier to ignite than any single component as methane and coal dust coexisted and resulted in the more serious consequences. Benedetto [15] pointed out that the lower explosive limit of coal dust could be obviously decreased after adding methane and its flame propagation velocity was increased. Ji [16] also studied the influence of methane content on the lower explosive limit and found that the value was evidently reduced and the explosion intensity was enhanced after adding methane. Zhao [17] also pointed out that the addition of methane could increase the explosion pressure of coal dust and its reaction rate was also enhanced with the increase of methane content. Chen [18] researched the effect of coal dust concentration on gas/dust mixture explosions and proposed that the temperature and propagation velocity of explosion flame showed a change trend of increase firstly and then decrease with the rise of coal dust content. Jing [19] also proposed that the flame propagation velocity showed the same variation law with the increase of coal dust content. And it was also pointed out that the higher volatile content would result in the enhancement of mixture explosions. Cloney [20] found that the particle size of coal dust had an important impact on the velocity history of mixture explosions. And it was predicted by the model that the velocity was significantly reduced as the coal dust particle size exceeded a critical value. Gao [21] found that the lower explosive limit could be reduced as the coal dust particle size decreased, and the explosion risk caused by coal dust was enhanced.

Meanwhile, scholars also studied the impact of working condition on gas/dust mixture explosions. Wang [22] pointed out that the ignition delay time had an evidently effect on mixture explosions and the explosion intensity showed a trend of increase firstly and then decrease with the rise of ignition delay time under 9.5% methane concentration. Deng [23] found that ignition energy had also an evidently influence on the mixture explosion pressures, and the influence of ignition energy was more obvious under the smaller energy. Besides, Zhou [24] studied the impact of obstacles on the flame propagation process of methane and coal dust mixture explosions, and found that obstacles could significantly accelerate the flame propagation and increase its propagation velocity inside a closed pipeline. And the obstacle with the sharper corner would cause more violent explosion.

Totally, the research on gas/dust mixture explosion characteristics had been conducted, and mainly focused on the explosion pressure and its influence factor [25,26]. Nevertheless, it is rare to research the change law and influence mechanism under multi-parameter conditions based on the combination of gas/dust mixture explosion parameter and flame propagation characteristics. Especially for the changes in the flame temperature and products [27]. Comprehensive analysis of flame propagation and explosion parameters could deeply reveal the gas/dust two-phase mixture explosion characteristics and its influence mechanism. Hence, the main purpose of this experiment was to study the parameter of methane and coal dust mixture explosions (the pressure peak ($\Delta P_{max}$), pressure rising rate ($dP/dt$), products and flame temperature ($T$)) and flame propagation characteristics (especially for the changes in flame morphology and its propagation velocity ($v$)) under multi-parameter operating conditions. According to the comprehensive analysis of explosion parameters, the characteristics of gas/dust mixture explosions and its influence mechanism can be analyzed in depth.

## 2. Experimental Setup

Figure 1 shows a schematic diagram of experiment apparatus for studying the characteristics and influence mechanism of gas/dust mixture explosions [28,29]. It mainly included eight subsystems. The main structure had been introduced in the

previous work [30], and only part of the structure was slightly modified. The powder injection system consisted of a Hartmann powder spraying device, a globe valve and a solenoid valve. The Hartmann powder spraying device was installed at the center of the bottom flange through a threaded connection [31]. The premix tank was sequentially connected with a globe valve and a solenoid valve by the bottom end tube bundle. The coal dust was placed in the groove of the Hartmann powder spraying device. A bottom-up vortex was formed as the high-speed premixed gas flow passed through the Hartmann powder spraying device from the bottom tube bundle due to the obstruction of the hemispherical "mushroom cap", lifting the dust in the groove and forming a dust cloud. The powder spraying method could achieve the relatively uniform dust cloud. The mixture gas of methane (Purity: 99.99%) and air was prepared inside the premixed tank by the partial pressure method and the determined vacuum degree inside the vessel was achieved by a vacuum pump. The powder injection pressure and time were determined by the calculation and ensured the methane concentration inside the vessel after power spraying. And the initial pressure was normal. The ignition system was composed of a high voltage converter, an AC contactor, a solid-state relay and an ignition electrode. The ignition electrode was placed at a central position with 110 mm height distance from the bottom flange (ignition voltage (U) = 8 kV). A high-speed camera (Photron V1212 (Japan)) was adopted to collect the flame propagation process of gas/dust mixture explosions. And flame propagation velocity history can be computed by the derivation of flame front position over time. To ensure the accuracy of flame propagation velocity, the acquisition frequency was set to 2000 fps. Namely, the change in flame front for 0.5 ms. A high-frequency pressure transmitter (Accuracy: 0.5% FS; Acquisition frequency: 200 Hz) and a high-speed thermocouple (S-type platinum rhodium; Range: 0–1800 °C) were installed at the center of vessel sidewall to record the pressure and flame temperature of mixture explosions [32]. The order of powder injection and ignition, as well as data collection, were realized through a program control and data collection system with a single channel of 2 M/s [33].

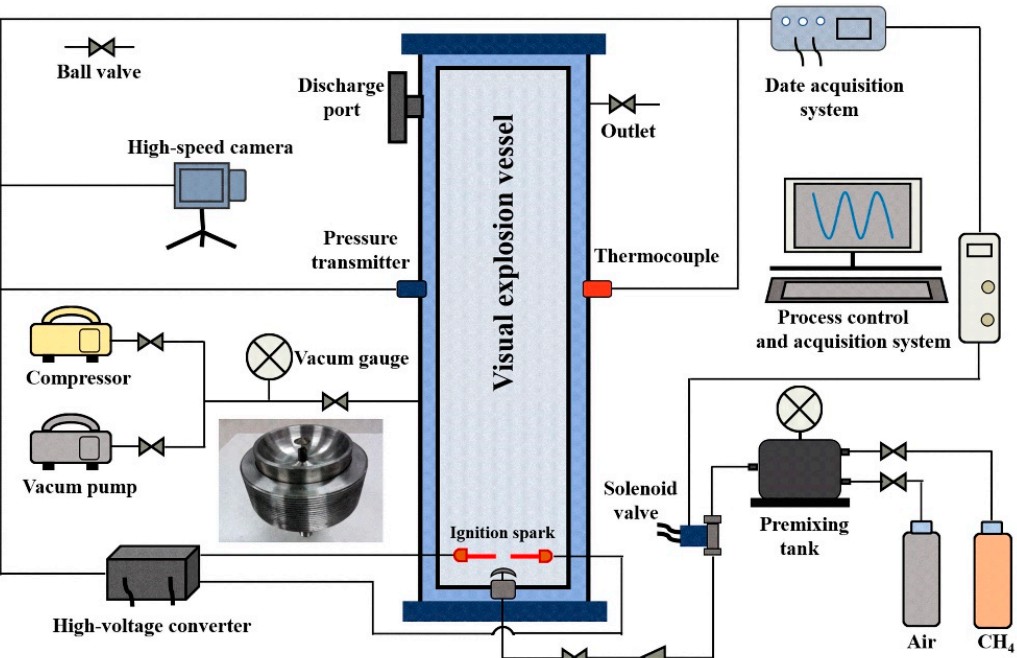

**Figure 1.** Schematic diagram of gas/dust mixture explosion experiment apparatus.

Before the experiment, the large coal block was pulverized by a grinder. And the coal dust with different particle size range was obtained by changing the screen mesh of different pore diameter. Three kinds of coal dust diameters ($d_{50}$ = 35.6 μm, 76.6 μm and 141.4 μm), five kinds of coal dust concentrations ($q$ = 100 g/m$^3$, 200 g/m$^3$, 300 g/m$^3$,

400 g/m$^3$ and 500 g/m$^3$) and five kinds of methane concentrations ($c$ = 2%, 4%, 6%, 8% and 10%) were selected for studying mixture explosion experiments. Figure 2 shows the columnar distribution and microstructure of three coal dust diameters. The morphology of the three types of coal dusts presented a uniform state and their diameter histories were approximately normal distribution [34]. Meanwhile, the component content of coal dust was measured by industrial analysis (M$_{ad}$ = 8.8%, A$_{ad}$ = 6.3%, V$_{ad}$ = 42.8%, FCad = 42.1%). Simultaneously, the pyrolysis characteristics of coal dust under the high temperature with a heating rate of 10 K/min was tested. It underwent three stages: the heat absorption, weight loss and stability. According to the comparison of explosion experiments under different powder injection pressures ($P_{st}$), the pressure was maximum as the $P_{st}$ was 0.40 MPa. Hence, 0.40 MPa was selected as the powder injection pressure during experiments. Each condition was repeated 3–4 times to guarantee the experimental accuracy. The data deviation was an important parameter characterizing experimental repeatability and it was calculated by the $\Delta P_{max}$ and the average value of repetitive experiments. During this experiment, the deviation value of $\Delta P_{max}$ was less than 5.5% through the data analysis.

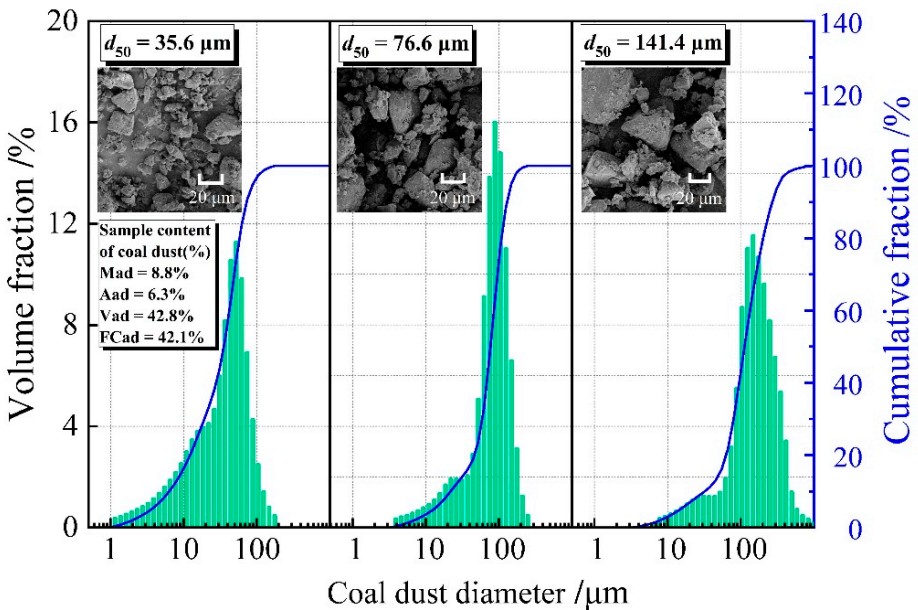

**Figure 2.** Columnar distribution of coal dust diameter.

## 3. Results and Discussion

### 3.1. Flame Propagation Characteristics

Figure 3 shows the comparison of mixture explosion flame propagation processes under five coal dust concentrations ($d_{50}$ = 76.6 μm; $c$ = 8%). The height of flame front at the same moment showed a change trend of increase firstly and then decrease as the coal dust concentration increased. As $q$ = 300 g/m$^3$, the flame front was the highest and brightest. The initial flame exhibited an irregular structure [35]. Meanwhile, the flame got dark and the irregular flame became more significant with the increase and decrease of coal dust concentration. As 100 g/m$^3$ < $q$ < 300 g/m$^3$, the quantity of coal dust per unit space was gradually increased. The heat released by methane explosion could be effectively absorbed by coal dust. Subsequently, the explosion intensity was increased due to the heat release from coal dust combustion. As the flame propagated, the flame front became the smooth and developed upward in an "elliptical" structure [36,37]. As $q$ = 300 g/m$^3$, the heat release of gas/dust two phase reactions after heat absorption resulted in that the explosion reaction was the most severe and the flame was the brightest. As 400 g/m$^3$ < $q$ < 500 g/m$^3$, the brightness was successively darkened and the flame began to become irregular. Especially the change was more significant as $q$ = 500 g/m$^3$. It was because that the reaction heat of methane explosion could be evidently aborted by the coal dust of higher concentration per

unit space. And the coal dust could not sufficiently occur the explosion reaction due to the limited oxygen concentration inside the confined space. Meanwhile, the combustion products could greatly consume the explosion reaction heat, leading to an evident reduction in flame brightness. And the flame front exhibited an evident wrinkle and instability. This explains that the dust concentration had a significant impact on the flame structure and its propagation characteristics of gas/dust mixture explosions [38].

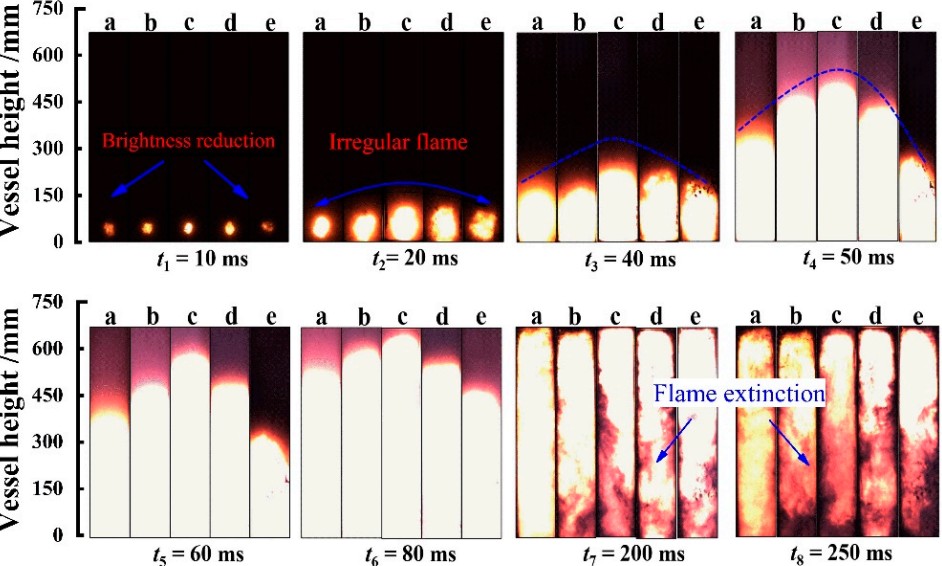

**Figure 3.** Comparison of flame propagation processes under five coal dust concentrations (a = 100 g/m$^3$, b = 200 g/m$^3$, c = 300 g/m$^3$, d = 400 g/m$^3$ and e = 500 g/m$^3$).

Figure 4 shows the comparison of velocity histories under different coal dust concentrations ($d_{50}$ = 76.6 μm; $c$ = 8%). The velocity history and its rising rate appeared a change trend of increase firstly and then decrease with the increase of coal dust concentration. As $q$ = 300 g/m$^3$, the corresponding velocity value was largest and the required time for reaching the top end was the shortest. The $v_{max}$ was evidently reduced from 17.7 to 12.9 m/s and 14.6 m/s with the change of coal dust concentration. Especially $v_{max}$ was the smallest as $q$ = 500 g/m$^3$. Figure 5 presents the effect of coal dust diameter on the velocity history of mixture explosions ($c$ = 8%; $q$ = 300 g/m$^3$). The velocity history was gradually reduced with the increase of coal dust diameter. In particular, $v_{max}$ was reduced from 20.1 to 17.1 m/s and its corresponding moment was also obviously delayed. It was because that the larger coal dust possessed a smaller specific surface area, further resulting in a relatively smaller contact area with oxygen and exhibited a slower explosion reaction rate and a darker flame structure [39].

Figure 6 illustrates the influence of methane concentration on the velocity history of mixture explosions ($d_{50}$ = 76.6 μm; $q$ = 300 g/m$^3$). As the methane concentration increased, the velocity history was also appeared an alteration trend of increase firstly and then decrease. Under 6% methane concentration, the velocity value was the maximal ($v_{max}$ = 21.5 m/s) and the corresponding moment of $v_{max}$ was the smallest. With the increase and decrease of methane concentration, the velocity history was evidently reduced. In particular, the velocity history was smallest ($v_{max}$ = 12.4 m/s) and the corresponding moment of $v_{max}$ was increased from 44 to 82 ms as $c$ = 10%. This was because that the methane and coal dust could not obtain sufficient oxygen during the explosion reaction as the methane concentration was increased to 10%, resulting in the inability of the heat released by the mixture system to maintain a high flame propagation velocity. Based on the Figures 4–6, the methane concentration and coal dust parameters could evidently affect the flame propagation characteristics of gas/dust two-phase mixture explosions.

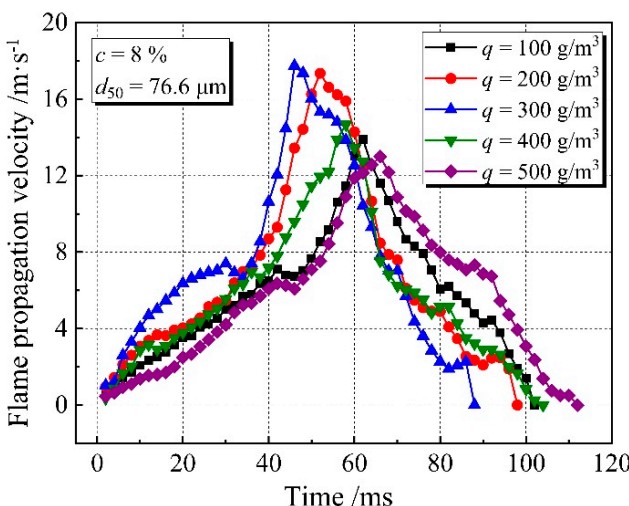

**Figure 4.** Comparison of velocity histories under five coal dust concentrations ($d_{50}$ = 76.6 μm; $c$ = 8%).

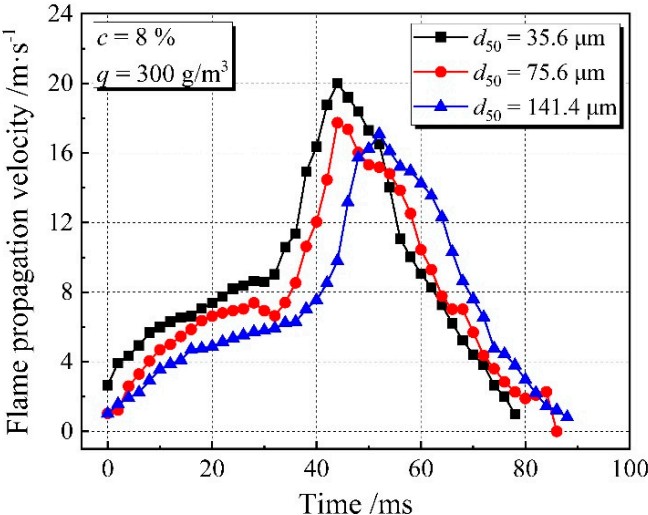

**Figure 5.** Comparison of velocity histories under three coal dust diameters ($c$ = 8%; $q$ = 300 g/m$^3$).

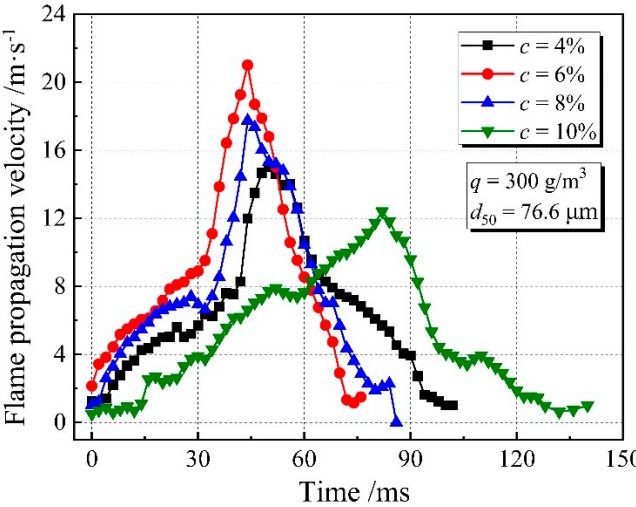

**Figure 6.** Comparison of velocity histories under five methane concentrations ($d_{50}$ = 76.6 μm; $q$ = 300 g/m$^3$).

*3.2. Explosion Pressure*

Figure 7 shows the comparison of mixture explosion pressure histories under five coal dust concentrations ($d_{50}$ = 76.6 μm; $c$ = 8%). The $\Delta P_{max}$ appeared a change trend of increase firstly and then decrease with the increase of coal dust concentration. The pressure history was maximal as $q$ = 300 g/m³ ($\Delta P_{max}$ = 0.61 MPa). With the increase and decrease of coal dust concentration, the pressure was significantly decreased. In particular, the $\Delta P_{max}$ was respectively reduced to 0.57 MPa and 0.52 MPa as $q$ = 100 g/m³ and 500 g/m³. Meanwhile, the concentration of coal dust corresponding to the maximum pressure was decreased in turn as the methane concentration was increased. As $c$ = 2% and 10%, the concentrations of coal dust corresponding to the maximum pressure were 500 g/m³ and 200 g/m³, respectively. Especially for 2% methane concentration, the mixture explosions would not occur as $q$ < 400 g/m³. This indicates that the coal dust content in the premixed gas of methane/air had a significant influence on the intensity of gas/dust two-phase mixture explosions.

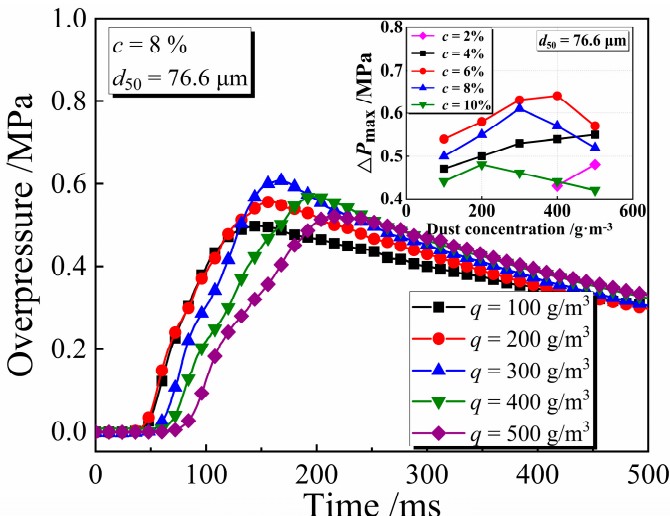

**Figure 7.** Comparison of pressure histories under different coal dust concentrations ($d_{50}$ = 76.6 μm; $c$ = 8%).

Figure 8 presents the comparison of mixture explosion pressure histories under three coal dust diameters ($q$ = 300 g/m³; $c$ = 8%). The pressure was evidently reduced as the coal dust diameter increased. In particular, $\Delta P_{max}$ was decreased by 3.17% and 11.11%, respectively. As $d_{50}$ = 141.4 μm, the pressure value was the smallest ($\Delta P_{max}$ = 0.56 MPa). Meanwhile, $\Delta P_{max}$ all appeared a decrease change trend as the coal dust diameter increased. Besides, the intensity of mixture explosions was maximal as $q$ = 300 g/m³. It also indicates that not only the concentration of coal dust could affect the explosion intensity, but also its specific surface area also could evidently affect the gas/dust two-phase mixture explosions. Figure 9 presents the comparison of mixture explosion pressure histories under different methane concentrations ($q$ = 300 g/m³; $d_{50}$ = 76.6 μm). The pressure history also showed a trend of increase firstly and then decrease as the methane concentration increased. As $c$ = 6%, $\Delta P_{max}$ was the maximal ($\Delta P_{max}$ = 0.63 MPa). With the increase and decrease of methane concentration, $\Delta P_{max}$ presented an obvious decrease trend. And the pressure all showed the same change trend under five coal dust concentrations. Especially for 2% methane concentration, the mixture explosions would not occur as $q$ < 400 g/m³. And the same phenomenon also occurred as $c$ > 10%. It explains that the concentrations of combustible gas and dust could evidently affect the explosive limit of gas/dust two phases.

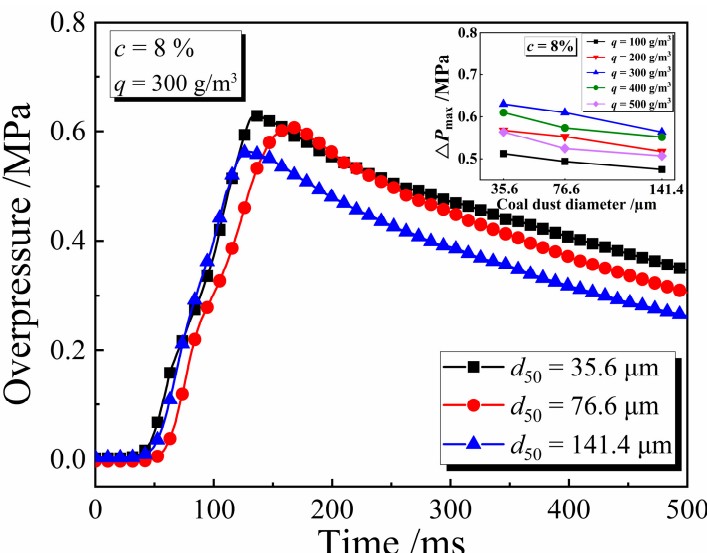

**Figure 8.** Comparison of pressure histories under different coal dust diameters ($c$ = 8%; $q$ = 300 g/m$^3$).

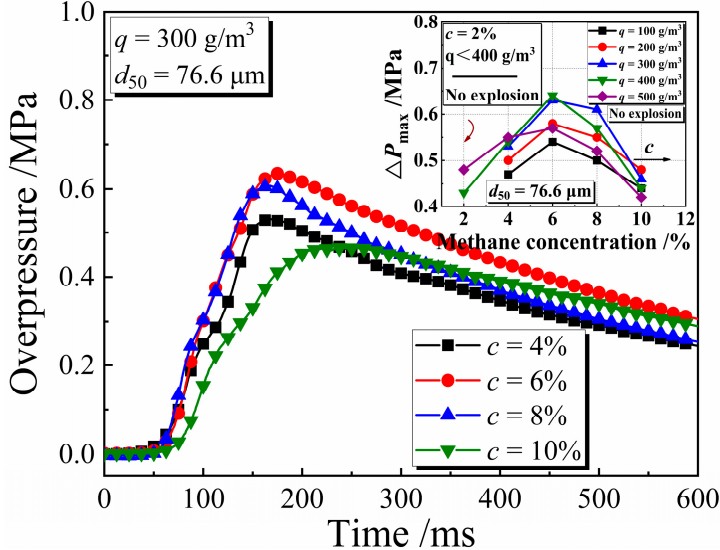

**Figure 9.** Comparison of pressure histories under different methane concentrations ($d_{50}$ = 76.6 μm; $q$ = 300 g/m$^3$).

### 3.3. Explosion Pressure Rising Rate

Figure 10 illustrates the comparison of ($dP/dt$) histories under five coal dust concentrations ($c$ = 8%; $d_{50}$ = 76.6 μm). It clear that the ($dP/dt$) history exhibited the same change trend as the pressure history. As the coal dust concentration increased, ($dP/dt$) history of mixture explosions also showed a change trend of increase firstly and then decrease. Meanwhile, the ($dP/dt$) history appeared two peaks (I and II peaks) and the I peak value was larger (($dP/dt$)$_{max}$). The two peak values were maximal as $q$ = 300 g/m$^3$. With the decrease and increase of coal dust concentration, the two peak values showed a decrease trend and the decrease degree of II peak value was more significant. Besides, the coal dust concentration corresponding to ($dP/dt$)$_{max}$ of mixture explosions was decreased sequentially as the methane concentration increased under the determined coal dust diameter. As $c$ = 4% and 10%, the corresponding coal dust concentrations were 500 g/m$^3$ and 200 g/m$^3$ (($dP/dt$)$_{max}$ = 8.4 MPa/s$^{-1}$ and 6.1 MPa/s$^{-1}$), respectively.

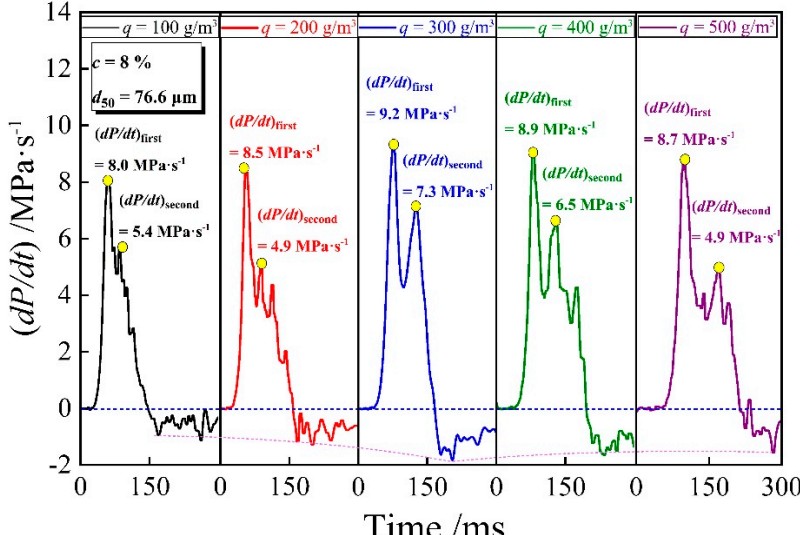

**Figure 10.** Comparison of (*dP/dt*) histories under different coal dust concentrations ($d_{50}$ = 76.6 μm; *c* = 8%).

Figure 11 presents the comparison of mixture explosion (*dP/dt*) histories under three coal dust diameters (*c* = 8%; *q* = 300 g/m³). As the coal dust diameter increased, the I and II peak values of (*dP/dt*) history presented an obvious decrease trend. In particular, the decrease extent of the II peak value was more significant and it was reduced by 30.39%. Meanwhile, (*dP/dt*)$_{max}$ all appeared a decrease trend and the maximum reduction extent reached 46.74% with the increase of coal dust diameter as *c* = 8%. Figure 12 shows the comparison of mixture explosion (*dP/dt*) histories under different methane concentrations (*q* = 300 g/m³; $d_{50}$ = 76.6 μm). It clear that the two peak values of (*dP/dt*) history showed a trend of increase firstly and then decrease as the methane concentration was raised. As *c* = 6%, the pressure rising rate was maximal. In addition, the (*dP/dt*)$_{max}$ were decreased by 25.49% and 33.33% under 4% and 6% methane concentrations. As $d_{50}$ = 76.6 μm, the pressure rising rates corresponding to 6% and 10% methane concentrations were the maximal and minimal values, respectively. The above explains that the reaction rate of gas/dust two-phase mixture explosions could be evidently affected by the methane content, coal dust diameter and its concentration, thereby affecting the explosion intensity.

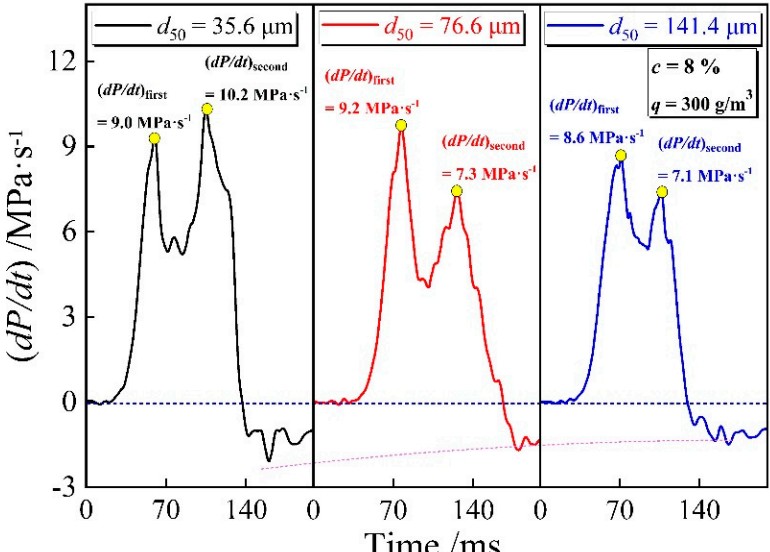

**Figure 11.** Comparison of (*dP/dt*) histories under different coal dust diameters (*c* = 8%; *q* = 300 g/m³).

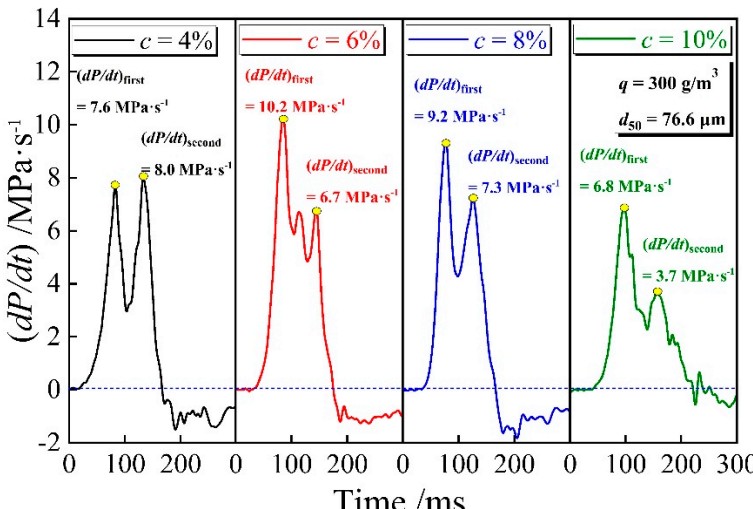

**Figure 12.** Comparison of $(dP/dt)$ histories under different methane concentrations ($d_{50}$ = 76.6 μm; $q$ = 300 g/m$^3$).

### 3.4. Flame Temperature

Figure 13 is the comparison of flame temperature histories under five coal dust concentrations ($c$ = 8%; $d_{50}$ = 76.6 μm). The temperature also showed the same change trend as the pressure history with the increase of coal dust concentration. Especially the flame temperature was the highest and $T_{max}$ reached 1687 °C as $q$ = 300 g/m$^3$. For 100 g/m$^3$ and 500 g/m$^3$ methane concentrations, $T_{max}$ were respectively reduced to 1315 °C and 1438 °C and their decrease extent were 22.05% and 14.76%. As $c$ = 8%, $T_{max}$ of mixture explosions all occurred at $q$ = 300 g/m$^3$ under three dust diameters. Meanwhile, $T_{max}$ presented an obvious decrease trend with the decrease and increase of coal dust concentration. Figure 14 presents the comparison of mixture explosion flame temperature histories under three coal dust diameters ($c$ = 8%; $q$ = 300 g/m$^3$). $T_{max}$ presented an obvious decrease trend (from 1781 to 1569 °C) and the slope of temperature history was also decreased as the coal dust diameter was increased. In addition, the influence of coal dust diameter on $T_{max}$ all showed a same change trend under five coal dust concentrations.

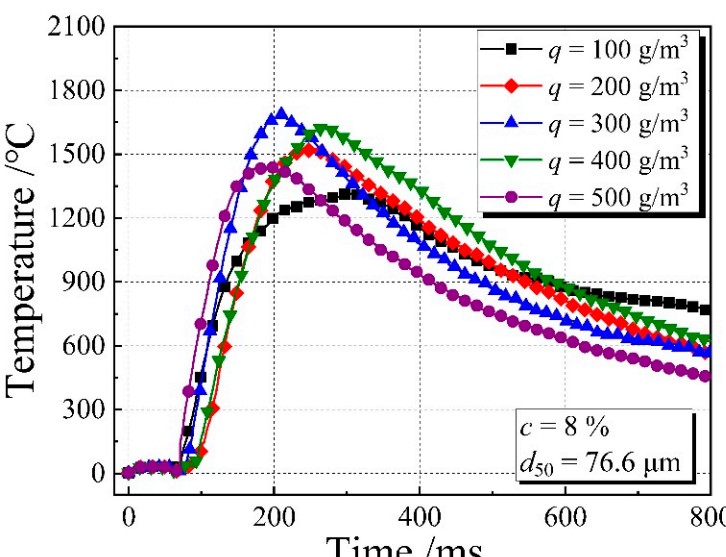

**Figure 13.** Comparison of temperature histories under different coal dust concentrations ($d_{50}$ = 76.6 μm; $c$ = 8%).

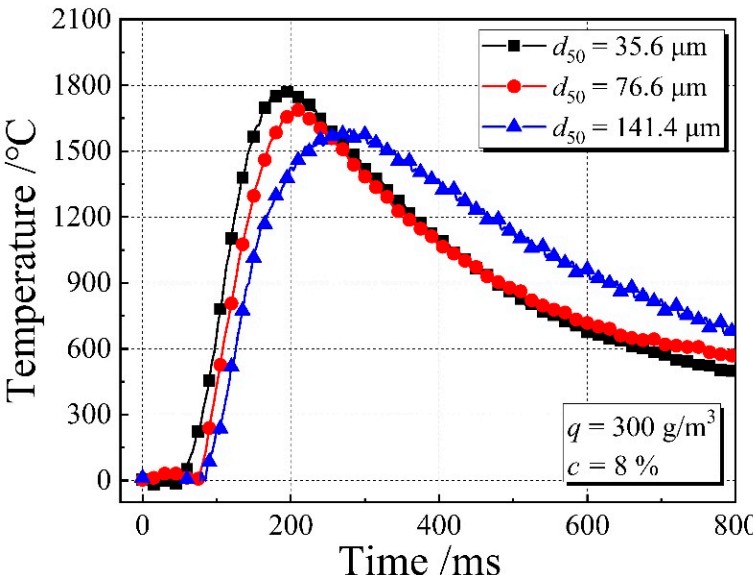

**Figure 14.** Comparison of temperature histories under different coal dust diameters (*c* = 8%; *q* = 300 g/m³).

Figure 15 presents the comparison of mixture explosion flame temperature histories under different methane concentrations (*q* = 300 g/m³; $d_{50}$ = 76.6 μm). $T_{max}$ of 6% methane concentration was the highest. With the change of methane concentration, the temperature history and its slope were decreased successively. Especially for 4% and 10% methane concentrations, $T_{max}$ was decreased by 22.19% and 44.85%, respectively. Besides, the coal dust concentration corresponding to the highest flame temperature was also successively reduced with the increase of methane concentration under 76.6 μm coal dust diameter. Especially, the coal dust concentration corresponding to the highest flame temperature was 200 g/m³ as *c* = 10%. This indicates that the combustion degree and heat release of gas/dust two-phase could be evidently affected by the combustion gas concentration, the dust diameter and its concentration.

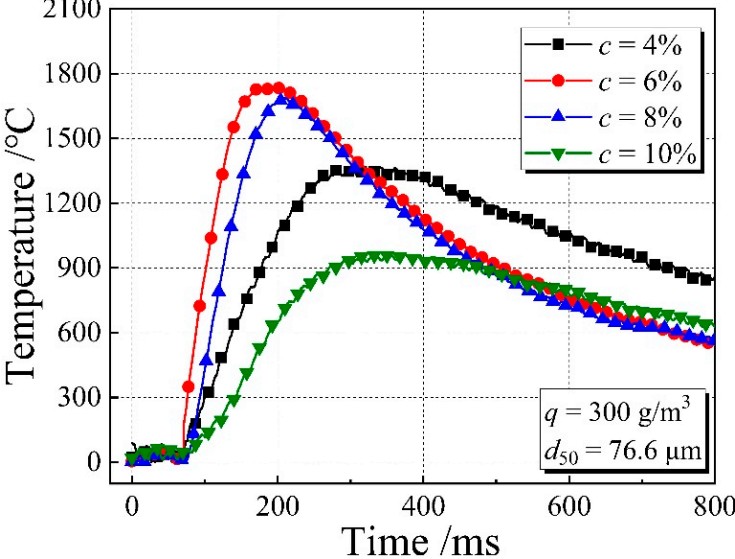

**Figure 15.** Comparison of temperature histories under different methane concentrations ($d_{50}$ = 76.6 μm; *q* = 300 g/m³).

## 4. Product Analysis

The DSC and TG histories of coal dust could intuitively reflect its thermal stability and decomposition oxidation characteristics after being heated. Figures 16 and 17 illustrate the TG and DSC histories of burned and unburned coal dusts in an oxygen atmosphere with the increase of temperature at a heating rate of 10 K/min, respectively [40,41]. The coal dust all presented three stages of the temperature rise, weight loss and stability after being heated, but there were the significant differences between the burned and unburned coal dust. It clear that from Figure 16, TG history of unburned coal dust occurred a slight change with the increase of temperature within the range of 0~263 °C. This was caused by the evaporation of moisture in coal dust after heating. At 263 °C, the coal dust occurred the pyrolysis. And TG history started to decline and the weight loss rate was gradually increased. The weight loss rate was the fastest and the corresponding DTG history reached a peak value (1.25%/°C) at 446 °C. This process was considered the acceleration weightlessness stage of coal dust. Meanwhile, the slope of coal dust DSC history was rapidly increased at this stage, indicating that the coal dust experienced a violent oxidation reaction and the heat release was continuously increased. During the whole thermal decomposition process (263 °C < *T* < 612 °C), there were two exothermic peaks on the DSC history at 446 °C and 484 °C, and the first peak (34.91 mW/mg) was significantly greater than the second peak (13.25 mW/mg). The appearance of the second exothermic peak resulted from the pyrolysis of stable aromatic rings inside the coal dust [42]. After the first peak value, the weight loss rate was rapidly decreased. As the temperature increased to 612 °C, the weight loss did not occur anymore and reached the thermal stability stage.

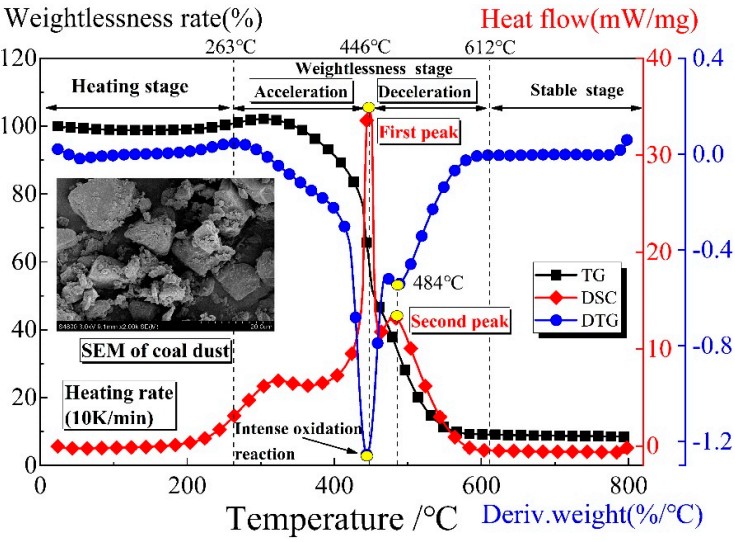

**Figure 16.** TG and DSC histories of unburned coal dust.

Compared with unburned coal dust, the thermal stability and decomposition oxidation of burned coal dust occurred an obvious change, as shown in Figure 17. At 274 °C, the burned coal dust appeared the pyrolysis and the corresponding temperature was slightly increased. After that, the coal dust entered the accelerated weight loss stage, but the weight loss rate was obviously decreased. Especially the peak value of weight loss rate (0.78%/°C) was evidently reduced and the corresponding temperature was increased (*T* = 479 °C). Meanwhile, the slope of DSC history during the accelerated weightlessness stage was also significantly decreased. Especially the exothermic peak value (18.18 mW/mg) of coal dust oxidation reaction was also significantly decreased and the corresponding temperature was evidently increased. Although the DSC history also appeared a small second peak, it was obviously reduced and the corresponding temperature was prolonged compared with the unburned coal dust. Besides, the temperature of coal dust reaching the stable stage

was also increased. It indicates that the content of stable aromatic rings in coal dust was significantly decreased after the combustion, but there was still a small pyrolysis reaction of remaining stable aromatic rings at the high temperature.

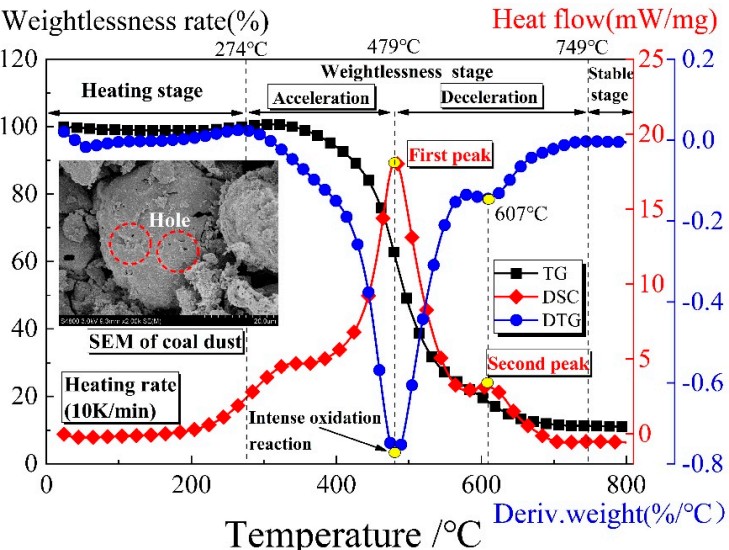

**Figure 17.** TG and DSC histories of burned coal dust.

The above indicates that coal dust participated in the gas/dust two-phase explosion reactions and the pyrolysis reaction of volatile matter resulted in an evident reduction in the weight loss and oxidation reaction rate of coal dust. It can be seen by the SEM that the unburned coal dust presented a blocky structure with sharp edges and corners [43]. Under the action of a high temperature flame, the volatile matter in coal dust continuously occurred the pyrolysis reaction. The macromolecular chemical bonds were decomposed into gaseous state ($CH_4$, CO, $CO_2$, $N_2$ and $O_2$, etc.) and small molecular liquids (hydrocarbons) [44], resulting in a continuous decrease in the functional groups involved in the oxidation reaction. Besides, the volatile matter gradually accumulated and expanded inside the coal dust, resulting in an obvious increase of coal dust volume [45]. Subsequently, the volatile substances were released from its surface after reaching a certain pressure, resulting in the numerous pore structure on the coal dust surface [46,47], as can be seen from Figure 17. The volatile gas generated by coal dust would promote the gas phase explosion reaction and heat transfer to the coal dust, further promoting the pyrolysis reaction of coal dust [48].

## 5. Conclusions

The characteristics and influence factor of gas/dust two-phase mixture explosions were studied experimentally. Based on the qualitative analysis of flame propagation, explosion intensity, flame temperature and products, the effects of methane and coal dust physical parameters on mixture explosion parameter were obtained. And the influence factor and its mechanism were analyzed deeply. Conclusions are summarized as follows:

First, the flame propagation of mixture explosions could be evidently affected by the coal dust parameters and methane concentration. The height of flame front at the same moment showed a change trend of increase firstly and then decrease with the rise of coal dust concentration. The $v_{\max}$ also presented the same change trend. In particular, the flame front was the highest and the brightness was the strongest as $q$ = 300 g/m$^3$. With the increase and decrease of coal dust concentration, the flame brightness was gradually darkened and the irregular flame front was more significant. The velocity history also appeared the same change trend as the methane concentration increased. And the maximal velocity value appeared at 6% methane concentration. However, the

time for reaching the vessel end was increased and $v_{\max}$ was gradually decreased as the coal dust dimeter increased.

Second, $\Delta P_{\max}$ all appeared a change trend of increase firstly and then decrease with the increases of methane and coal dust concentrations. The coal dust concentration corresponding to the maximum pressure existed difference and was decreased in turn with the rise of methane concentration. As $c = 2\%$ and $10\%$, the corresponding coal dust concentrations were 500 g/m$^3$ and 200 g/m$^3$, respectively. However, $\Delta P_{\max}$ was obviously reduced with the rise of coal dust diameter. The (*dP/dt*) history also showed the similar changing trend to the pressure with the changes of methane and coal dust concentrations and appeared two peak values. With the rise of coal dust diameter, the two peak values all showed a decrease trend. And the reduction extent of second peak value was more significant.

Third, $T_{\max}$ presented the similar changing trend to the pressure with the increases of methane and coal dust concentrations (increase firstly and then decrease). However, $T_{\max}$ showed a decreasing change trend as the coal dust diameter raised. Besides, the coal dust participated in the gas/dust two-phase explosion reaction at high temperature. The thermal stability and decomposition oxidation characteristics of burned coal dust appeared an obvious change compared with the unburned coal dust. The pyrolysis reaction of volatile matter led to an obvious reduction in the weight loss and oxidation reaction rate. And the precipitation of volatile matter also resulted in an obvious pore structure on its surface.

For better prevention of gas/dust two-phase mixture explosions, three suggestions are proposed: (1) The coal dust concentration corresponding to the maximum pressure was different with the change of methane concentration. In practice, the mixture concentration should be kept away from the concentration corresponding to the maximum mixture explosion pressures. (2) Enterprises involved in powder production should avoid the presence of small size dust as much as possible and the mixture of the two should exceed its explosion limit. (3) The space where dust existed should be sprayed with fine water mist to decrease the precipitation of volatile matter inside the dust and its combustion rate.

**Author Contributions:** Conceptualization, Y.W. and H.W.; methodology, Z.W. and X.C.; investigation, X.C.; writing—original draft preparation, Z.W. and X.C.; writing—review and editing, Z.W. and X.C.; funding acquisition, Y.W. and X.C.; All authors have read and agreed to the published version of the manuscript.

**Funding:** This research was funded by National project funding for National Natural Science Foundation of China (Project No. 52274210; 21865036; 51806056), the China Postdoctoral Science Foundation (Projects No. 2020M681573), the Natural Science Research Projects of Jiangsu Province Universities (Projects No. 20KJA620001), the Postdoctoral Research Funding Program of Jiangsu Province, State Key Laboratory of Coal Resources and Safe Mining-Xinjiang Institute Engineering Joint Open Research Fund Project (No. SKLCRSM-XJIEKF007).

**Institutional Review Board Statement:** Not applicable.

**Informed Consent Statement:** Not applicable.

**Data Availability Statement:** Not applicable.

**Acknowledgments:** We thank the National project funding for National Natural Science Foundation of China (Project No. 52274210; 21865036; 51806056), the China Postdoctoral Science Foundation (Projects No. 2020M681573), the Natural Science Research Projects of Jiangsu Province Universities (Projects No. 20KJA620001), the Postdoctoral Research Funding Program of Jiangsu Province, State Key Laboratory of Coal Resources and Safe Mining-Xinjiang Institute Engineering Joint Open Research Fund Project (No. SKLCRSM-XJIEKF007).

**Conflicts of Interest:** The authors declare no conflict of interest.

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
