# Peer review of "Study on the Characteristics and Influence Factor of Methane and Coal Dust Gas/Solid Two-Phase Mixture Explosions"

_fire, doi:10.3390/fire6090359_

Round 1
Reviewer 1 Report
· The abstract is comprehensive but lacking of the conclusion as the last sentence.
· Line 75-77 as the problem statement should be cited.
· Also why only explosion pressure and its influence factor by the current study? Can you elaborate the problem statement further.
· Your experimental procedure seems common and if you refer elsewhere, please include citation.
· Do you have statistical analysis to be reported?
· Some data seems without the experimental procedure. Please elaborate the procedures for all the reported data
· Overall, the paper is very comprehensive and minor correction is required.
Author Response
Response to reviewer#1:
Comment (1): The abstract is comprehensive but lacking of the conclusion as the last sentence.
Response (1): The conclusion as the last sentence in the abstract has been added.
Please see the last sentence of abstract in the revised manuscript.
“The physical parameters and internal components of coal dust were important factors affecting the reaction rates of gas/dust mixture explosions.”.
Comment (2): Line 75-77 as the problem statement should be cited.
Response (2): The relevant literature has been cited in the problem statement.
Please see the corresponding parts in the revised manuscript.
Reference:
25. Yu, X.Z., Zhang, Z.H., Yan, X.Q., Wang, Z., Yu, J.L., Gao, W. Explosion characteristics and combustion mechanism of hydrogen/tungsten dust hybrid mixtures. Fuel 2023, 332, 126017.
26. Ji, W.T., Gan, X,Y., Li, L., Li, Z., Wen, X.P., Wang, Y. Prediction of the explosion severity of hybrid mixtures. Powder Technol. 2022, 400, 117273.
27. Xiong, X.X., Gao, K., Mu, J., Li, B., Zhang, D., Xie, L.F. Study on explosion characteristic parameters and induction mechanism of magnesium powder/hydrogen hybrids. Fuel 2022, 326, 125077.
Comment (3): Also why only explosion pressure and its influence factor by the current study? Can you elaborate the problem statement further?
Response (3): Gas/dust mixture explosions were the process in which two-phase substances undergo the rapid chemical reactions and generate the high temperature and pressure. The explosion pressure and temperature are the main parameters to measure the explosions characteristics and the main factors to cause the harm. Meanwhile, the gas/dust explosion process was affected by multiple factors. However, the gas concentration and coal dust physical parameters (the dust concentration and diameter) are the most important influencing factors, and are also important parameters for measuring the risk of two-phase explosions. Therefore, this experiment comprehensively analyzed the flame propagation characteristics, explosion pressure and its rising rate, and temperature changes.
Please see the Introduction in the revised manuscript.
Comment (4): Your experimental procedure seems common and if you refer elsewhere, please include citation.
Response (4): In the preliminary work, this experimental procedure was used to study the explosion characteristics. According to expert suggestions, the relevant literature has been cited.
Please see the revised manuscript.
Reference:
28. Cao, X. Y., Bi, M.S., Ren, J.J., Chen, B. Experimental research on explosion suppression affected by ultrafine water mist containing different additives. J Hazard Mater 2019, 368, 613-620.
29. Cao, X.Y., Ren, J.J., Zhou, Y.H., Wang, Q.J., Gao, X.L., Bi, M.S. Suppression of methane/air explosion by ultrafine water mist containing sodium chloride additive. J Hazard Mater 2015, 285, 311-318.
Comment (5): Do you have statistical analysis to be reported?
Response (5): Each condition was repeated 3 - 4 times to guarantee the experimental accuracy. For the deviation value of â–³Pmax, the statistical analysis was conducted. The data deviation was an important parameter characterizing experimental repeatability and it was calculated by the â–³Pmax and the average value of repetitive experiments. During this experiment, the deviation value of â–³Pmax was less than 5.5% through the data analysis.
Please see the paragraph 2 (line 16-20) of Experimental setup.
Comment (6): Some data seems without the experimental procedure. Please elaborate the procedures for all the reported data?
Response (6): The procedures for all the reported data has been elaborated in the revised manuscript.
Please see the revised manuscript.
“Before the experiment, the large coal block was pulverized by a grinder. And the coal dust with different particle size range was obtained by changing the screen mesh of different pore diameter.”
“Meanwhile, the component content of coal dust was measured by industrial analysis (Mad = 8.8%, Aad = 6.3%, Vad = 42.8%, FCad = 42.1%).”
“A high-speed camera (Photron V1212) was adopted to collect the flame propagation process of gas/dust mixture explosions. And flame propagation velocity history can be computed by the derivation of flame front position over time. To ensure the accuracy of flame propagation velocity, the acquisition frequency was set to 2000 fps. Namely, the change in flame front for 0.5 ms.”
“A high-frequency pressure transmitter (Accuracy: 0.5% FS; Acquisition frequency: 200 Hz) and a high-speed thermocouple (S-type platinum rhodium; Range: 0 - 1800 ℃) were installed at the center of vessel sidewall to record the pressure and flame temperature of mixture explosions”.

Reviewer 2 Report
The paper presents an excellent experimental test results on the characteristics of methane and coal dust two phase system hybrid explosions inside a closed vessel. It shows a good innovation and theory in the change regular of explosion parameters and analysis of coal dust products under different influence factors. Simultaneously, the influence laws of coal dust parameter and methane concentration also were obtained. This is really an interesting and valuable work, however, some following questions should be stressed before it can be accepted:
(1) The physical properties of coal dust can obviously affect its explosive reaction rate and should be further explained in the experiment part.
(2) The methane concentrations of five premixed gases were selected during mixture explosions (c = 2%, 4%, 6%, 8% and 10%), why only the concentration with below 10% was selected during the experiment?
(3) To ensure the accuracy of experimental data, the physical parameters of dust should be further introduced.
(4) In order to ensure the accuracy of flame propagation velocity, its calculation method should be explained in detail.
(5) Some expressions in the manuscript need be further improved. Please read the manuscript carefully and make modifications.
Minor editing of English language required.
Author Response
Response to reviewer#2:
Reviewer 2: The paper presents an excellent experimental test results on the characteristics of methane and coal dust two phase system hybrid explosions inside a closed vessel. It shows a good innovation and theory in the change regular of explosion parameters and analysis of coal dust products under different influence factors. Simultaneously, the influence laws of coal dust parameter and methane concentration also were obtained. This is really an interesting and valuable work, however, some following questions should be stressed before it can be accepted:
Comment (1): The physical properties of coal dust can obviously affect its explosive reaction rate and should be further explained in the experiment part.
Response (1): The physical properties of coal dust has been added in the experiment part.
Please see the paragraph 2 (line 1-6) and (line 9-13) of Experimental setup.
“Before the experiment, the large coal block was pulverized by a grinder. And the coal dust with different particle size range was obtained by changing the screen mesh of different pore diameter. Three kinds of coal dust diameters (d50 = 35.6 μm, 76.6 μm and 141.4 μm), five kinds of coal dust concentrations (q = 100 g/m3, 200 g/m3, 300 g/m3, 400 g/m3 and 500 g/m3) and five kinds of methane concentrations (c = 2%, 4%, 6%, 8% and 10%) were selected for studying mixture explosion experiments.”
“The component content of coal dust was measured by industrial analysis (Mad = 8.8%, Aad = 6.3%, Vad = 42.8%, FCad = 42.1%). Simultaneously, the pyrolysis characteristics of coal dust under the high temperature with a heating rate of 10 K/min was tested. It underwent three stages: the heat absorption, weight loss and stability.”
Comment (2): The methane concentrations of five premixed gases were selected during mixture explosions (c = 2%, 4%, 6%, 8% and 10%), why only the concentration with below 10% was selected during the experiment?
Response (2): The explosive limit of methane is 4.5%-15%. However, the explosive limit of methane can be evidently affected after adding coal dust. Through experiments, it was found that the methane/coal dust mixture explosions would not occur as c = 2% and q < 400 g/m3. Simultaneously, the mixture explosion would also not occur as c > 10%. Therefore, the effective comparative analysis of mixture explosions cannot be conducted. Hence, the methane concentration with c ≤ 10% and c > 2% was selected during the experiment to achieve effective comparative analysis.
Comment (3): To ensure the accuracy of experimental data, the physical parameters of dust should be further introduced.
Response (3): According to reviewer’s suggestion, the physical parameter of dust has been further introduced.
Please see the Product analysis part in the revised manuscript.
“Before the experiment, the large coal block was pulverized by a grinder. And the coal dust with different particle size range was obtained by changing the screen mesh of different pore diameter. Three kinds of coal dust diameters (d50 = 35.6 μm, 76.6 μm and 141.4 μm), five kinds of coal dust concentrations (q = 100 g/m3, 200 g/m3, 300 g/m3, 400 g/m3 and 500 g/m3) and five kinds of methane concentrations (c = 2%, 4%, 6%, 8% and 10%) were selected for studying mixture explosion experiments.
The component content of coal dust was measured by the industrial analysis (Mad = 8.5%; Aad = 6.3; Vad = 42.8%; FCad = 42.0%). Simultaneously, the pyrolysis characteristics of coal dust under the high temperature with a heating rate of 10 K/min was tested. It underwent three stages: the heat absorption, weight loss and stability. The coal dust all presented three stages of the temperature rise, weight loss and stability after being heated, but there were the significant differences between the burned and unburned coal dust. Before 263 ℃, the coal dust exhibited an endothermic process. Thereafter, the pyrolysis occurred and the weight loss rate was gradually increased. At 446 ℃, the weight loss rate was the fastest and its corresponding DTG history reached the peak value. Subsequently, the weight loss rate was evidently reduced. After reaching 612 ℃, the weight loss of coal dust reached a stable state.”
Comment (4): In order to ensure the accuracy of flame propagation velocity, its calculation method should be explained in detail.
Response (4): A high-speed camera (Photron V1212) was adopted to collect the flame propagation process of gas/dust mixture explosions. And flame propagation velocity history can be computed by the derivation of flame front position over time. To ensure the accuracy of flame propagation velocity, the acquisition frequency was set to 2000 fps. Namely, the change in flame front for 0.5 ms.
Please see the paragraph 1 (line 21-26) of Experimental setup.
Comment (5): Some expressions in the manuscript need be further improved. Please read the manuscript carefully and make modifications.
Response (5): The language has been edited by professional retouching agency.
Please see the revised manuscript.

Reviewer 3 Report
This article focused on an experimental study on the characteristics and influence factor of gas/dust two-phase mixture explosions. Based on the qualitative analysis of flame propagation, explosion intensity, flame temperature and products, the effects of methane and coal dust physical parameters on mixture explosion parameter were obtained. The research is potentially novel and worthy of eventual publication. Overall, I think this article is very suitable for publication after addressing the following concerns.
 There are many factors that affect the explosion of gas/dust two-phase explosions, how to ensure the accuracy and repeatability of experiments.
 Current research on gas and dust explosions has been conducted, and some relevant literature should be added to this paper to better describe novelty.
 The physical properties of coal powder are an important influencing factor on its explosion rate, which should be introduced and explained in this paper.
 To ensure the accuracy of the collected experimental data, the accuracy of the corresponding collection equipment should be further introduced.
 The language expression of the article and use of the grammar need further improvement. I do suggest to have some external language editing done by the professional.
The English should be further improved.
Author Response
Response to reviewer#3:
Reviewer 3: This article focused on an experimental study on the characteristics and influence factor of gas/dust two-phase mixture explosions. Based on the qualitative analysis of flame propagation, explosion intensity, flame temperature and products, the effects of methane and coal dust physical parameters on mixture explosion parameter were obtained. The research is potentially novel and worthy of eventual publication. Overall, I think this article is very suitable for publication after addressing the following concerns.
Comment (1): There are many factors that affect the explosion of gas/dust two-phase explosions, how to ensure the accuracy and repeatability of experiments.
Response (1): To ensure the accuracy and repeatability of experiments, two aspects of testing equipment and experimental methods were considered.
First, to ensure the accuracy and effectiveness of experimental data, the physical parameter of coal dust was firstly determined before the experiments. For example, the coal dust component was tested and the morphological structure (× 2000) was analyzed. Its diameter distribution was also tested by a laser particle size analyzer and the pyrolysis characteristics of coal dust under the high temperature with a heating rate of 10 K/min was tested. Simultaneously, the mist diameter and atomization rate were also tested.
Second, the instruments and equipment used for data collection should ensure its accuracy. For instance, the pressure was collected using a high-frequency dynamic pressure transmitter (range: 0-2.5 MPa; response frequency: 200 kHz). The temperature change was recorded using an S-type platinum rhodium thermocouple (range: 0-1600 ℃). A high-speed camera (V1212) was used to gather the changes in flame morphology (2000 fps). An eight-channel high frequency data acquisition card (USB2871; the acquisition frequency: 200 kHz) was adopted to realize the program control and data acquisition. Furthermore, the product changes of coal dust before and after explosion were tested by an infrared spectrometer. Through the programming language, the orders of the image acquisition, the dust spraying, the explosion pressure and temperature data acquisitions, the ignition were achieved and their adjacent interval time was determined.
Besides, the deviation value of ΔPmax was less than 5.5% through the repeated experiments, indicating the good repeatability and reliability during experiment.
Please see the Experimental setup part.
Comment (2): Current research on gas and dust explosions has been conducted, and some relevant literature should be added to this paper to better describe novelty.
Response (2): Thank you for the expert's advice, the related literature has been added in this paper.
Reference:
25. Yu, X.Z., Zhang, Z.H., Yan, X.Q., Wang, Z., Yu, J.L., Gao, W. Explosion characteristics and combustion mechanism of hydrogen/tungsten dust hybrid mixtures. Fuel 2023, 332, 126017.
26. Ji, W.T., Gan, X,Y., Li, L., Li, Z., Wen, X.P., Wang, Y. Prediction of the explosion severity of hybrid mixtures. Powder Technol. 2022, 400, 117273.
27. Xiong, X.X., Gao, K., Mu, J., Li, B., Zhang, D., Xie, L.F. Study on explosion characteristic parameters and induction mechanism of magnesium powder/hydrogen hybrids. Fuel 2022, 326, 125077.
Comment (3): The physical properties of coal powder are an important influencing factor on its explosion rate, which should be introduced and explained in this paper.
Response (3): According to reviewer’s suggestion, the physical parameter of dust has been further introduced.
Please see the Experiments and Product analysis parts in the revised manuscript.
“Before the experiment, the large coal block was pulverized by a grinder. And the coal dust with different particle size range was obtained by changing the screen mesh of different pore diameter. Three kinds of coal dust diameters (d50 = 35.6 μm, 76.6 μm and 141.4 μm), five kinds of coal dust concentrations (q = 100 g/m3, 200 g/m3, 300 g/m3, 400 g/m3 and 500 g/m3) and five kinds of methane concentrations (c = 2%, 4%, 6%, 8% and 10%) were selected for studying mixture explosion experiments.
The component content of coal dust was measured by the industrial analysis (Mad = 8.5%; Aad = 6.3; Vad = 42.8%; FCad = 42.0%). Simultaneously, the pyrolysis characteristics of coal dust under the high temperature with a heating rate of 10 K/min was tested. It underwent three stages: the heat absorption, weight loss and stability. The coal dust all presented three stages of the temperature rise, weight loss and stability after being heated, but there were the significant differences between the burned and unburned coal dust. Before 263 ℃, the coal dust exhibited an endothermic process. Thereafter, the pyrolysis occurred and the weight loss rate was gradually increased. At 446 ℃, the weight loss rate was the fastest and its corresponding DTG history reached the peak value. Subsequently, the weight loss rate was evidently reduced. After reaching 612 ℃, the weight loss of coal dust reached a stable state.”
Comment (4): To ensure the accuracy of the collected experimental data, the accuracy of the corresponding collection equipment should be further introduced.
Response (4): According to the review’s suggestion, the accuracy of the corresponding collection equipment has been further introduced.
Please see the paragraph 1 (line 21-29) of Experimental setup.
“A high-speed camera (Photron V1212) was adopted to collect the flame propagation process of gas/dust mixture explosions. And flame propagation velocity history can be computed by the derivation of flame front position over time. To ensure the accuracy of flame propagation velocity, the acquisition frequency was set to 2000 fps. Namely, the change in flame front for 0.5 ms. A high-frequency pressure transmitter (Accuracy: 0.5% FS; Acquisition frequency: 200 Hz) and a high-speed thermocouple (S-type platinum rhodium; Range: 0 - 1800 ℃) were installed at the center of vessel sidewall to record the pressure and flame temperature of mixture explosions [32]. The order of powder injection and ignition, as well as data collection, were realized through a program control and data collection system with a single channel of 2 M/s [33]”
Comment (5): The language expression of the article and use of the grammar need further improvement. I do suggest to have some external language editing done by the professional.
Response (5): According to the review’s suggestion, the language has been edited by professional retouching agency.
Please see the revised manuscript.
